# Joint application of metagenomic next-generation sequencing and histopathological examination for the diagnosis of pulmonary infectious disease

Linhui Yang,[1] Kaige Wang,[1] Yalun Li,[1] Weimin Li,[1] Dan Liu[1]

**ABSTRACT**   The diagnosis of some pulmonary infectious diseases and their pathogens is very difficult. We clarified the diagnostic value of the joint application of metagenomic next-generation sequencing (mNGS) and histopathological examination for the diagnosis of pulmonary infectious diseases in this study. We retrospectively enrolled 108 patients with infectious and noninfectious diseases who visited the West China Hospital of Sichuan University from January 2019 to June 2022. Pathogens in core needle biopsy tissue were detected using mNGS and traditional pathological examination. The diagnostic accuracy of the combined mNGS and histopathology protocol for pulmonary infectious diseases was contrasted with that of histopathological examination alone. The area under the receiver operating characteristic curve (ROC curve) of mNGS combined with histopathological examination was 0.840 for pulmonary infectious diseases, and the area under the ROC curve of histopathological examination alone was 0.644. The areas under the ROC curve of mNGS combined with histopathological examination for pulmonary fungal infection, pulmonary tuberculosis, and lung abscess were 0.876, 0.875, and 0.840, respectively. In pulmonary infectious diseases, the areas under the ROC curve of histopathological examination alone for the above three types of pulmonary infections were 0.760, 0.719, and 0.575, respectively. For pulmonary infectious diseases, a joint application of metagenomic next-generation sequencing and histopathological examination can not only be used for differential diagnosis but also increase the positive rate of pathogen identification compared with histopathological examination alone, especially for rare pathogen infections.

**IMPORTANCE**  The diagnosis of some pulmonary infectious diseases and their pathogens is very difficult. A more precise diagnosis of pulmonary infectious diseases can help clinicians use proper antibiotics as well as reduce the development of drug-resistant bacteria. In this study, we performed both mNGS and pathology on lung puncture biopsy tissue from patients and found that combined mNGS and histopathology testing was significantly more effective than histopathology testing alone in detecting infectious diseases and identifying infectious diseases. In addition, the combined approach improves the detection rate of pathogenic microorganisms in infectious diseases and can be used to guide precision clinical treatment.

**KEYWORDS**    histopathology, mNGS, microbiology, pulmonary, infectious disease

Currently, pulmonary infection remains an important cause of morbidity and mortality worldwide (1, 2). Timely diagnosis and accurate identification of pathogens play an important role in the precise application of anti-infective drugs, shortening the course of the disease, improving the prognosis of patients, and reducing the generation of drug-resistant pathogenic microorganisms. With the advancement and

Address correspondence to Dan Liu, liudan10965@wchscu.cn, or Weimin Li, weimi003@yahoo.com.

Linhui Yang and Kaige Wang contributed equally to this article. The order of authorship of this article was jointly decided after discussion among all the authors.

The authors declare no conflict of interest.

See the funding table on p. 11.

diversification of clinical testing methods, there are now many techniques, such as tissue slices, tissue culture, polymerase chain reaction (PCR), and special chemical staining, that can help clinicians more accurately determine the nature of lung lesions. However, the culture of pathogenic microorganisms is time-consuming, the positive rate is about 27%–40% (3, 4), and many patients with pulmonary infection are often missed. In addition, detection methods such as PCR and special staining can only detect several specific pathogenic microorganisms, which cannot meet the needs of clinical detection. Moreover, the pathogens that cause community-acquired pneumonia are diverse, and clinical care is often initiated with empirical anti-infective therapy without identifying a clear pathogen, leading to an increase in drug-resistant bacteria (5).

In recent years, the advantages of metagenomic next-generation sequencing (mNGS) in infectious diseases have attracted increasing attention. It takes less time and enables unbiased analysis of a large number of pathogenic microorganisms (6, 7), including bacteria, fungi, and viruses. Currently, mNGS is commonly used in the detection of pathogenic bacteria in the blood and bronchoalveolar lavage fluid (BALF), and a large number of studies have reported that mNGS is more efficient than culture, PCR, and special staining methods (6). However, there are few studies on the combination of mNGS and computed tomography (CT)-guided percutaneous pulmonary puncture using lung biopsy tissue to detect microorganisms.

In this study, we retrospectively analyzed the clinical data of patients who had completed percutaneous lung puncture and tissue mNGS detection in our hospital in the past two years. At the same time, we compared the diagnostic efficacy of combined use of tissue mNGS and histopathology with that of histopathology alone for pulmonary infectious diseases.

## MATERIALS AND METHODS

### Patients and samples

A total of 4,032 patients who visited the West China Hospital of Sichuan University from January 2019 to June 2022 were retrospectively analyzed. These patients cannot be clearly diagnosed by noninvasive examination. All patients had undergone a CT-guided percutaneous lung biopsy on the advice of a senior physician. One hundred eighty patients whose lung lesions were considered to be caused by specific pathogenic microbial infections but required further puncture examination and who simultaneously had their core needle biopsy tissue examined by mNGS were included in the study. The exclusion criteria for this study included (i) patients whose core needle biopsy tissue had no traceable mNGS results and (ii) patients who lacked significant clinical information. The process for screening patients is shown in Fig. 1. This study was approved by the Ethics Committee of West China Hospital, Sichuan University. We finally included 108 patients and collected some relevant clinical information in the electronic medical record system, such as age, sex, disease duration, symptoms, past medical history, laboratory findings, pathological histological results, tissue culture results, mNGS results, imaging manifestations, treatment options, and disease progression.

### Sample processing and analysis

Our hospital has established a precision medicine center that can be dedicated to mNGS testing of patients' specimens. In short, when a patient undergoes a percutaneous lung puncture, the operator takes two pieces of core needle biopsy tissue of approximately equal length under real-time CT guidance and stores one piece in buffered 10% formalin in water for histopathological testing and one piece in a dry, sterile tube for mNGS testing. The detection process of mNGS can be roughly divided into five steps: nucleic acid extraction, library construction, sequencing, raw signal analysis, and report interpretation. The specific process is as follows: DNA was isolated using magnetic bead adsorption after lung tissue samples were forcibly fractured with a wall breaker and

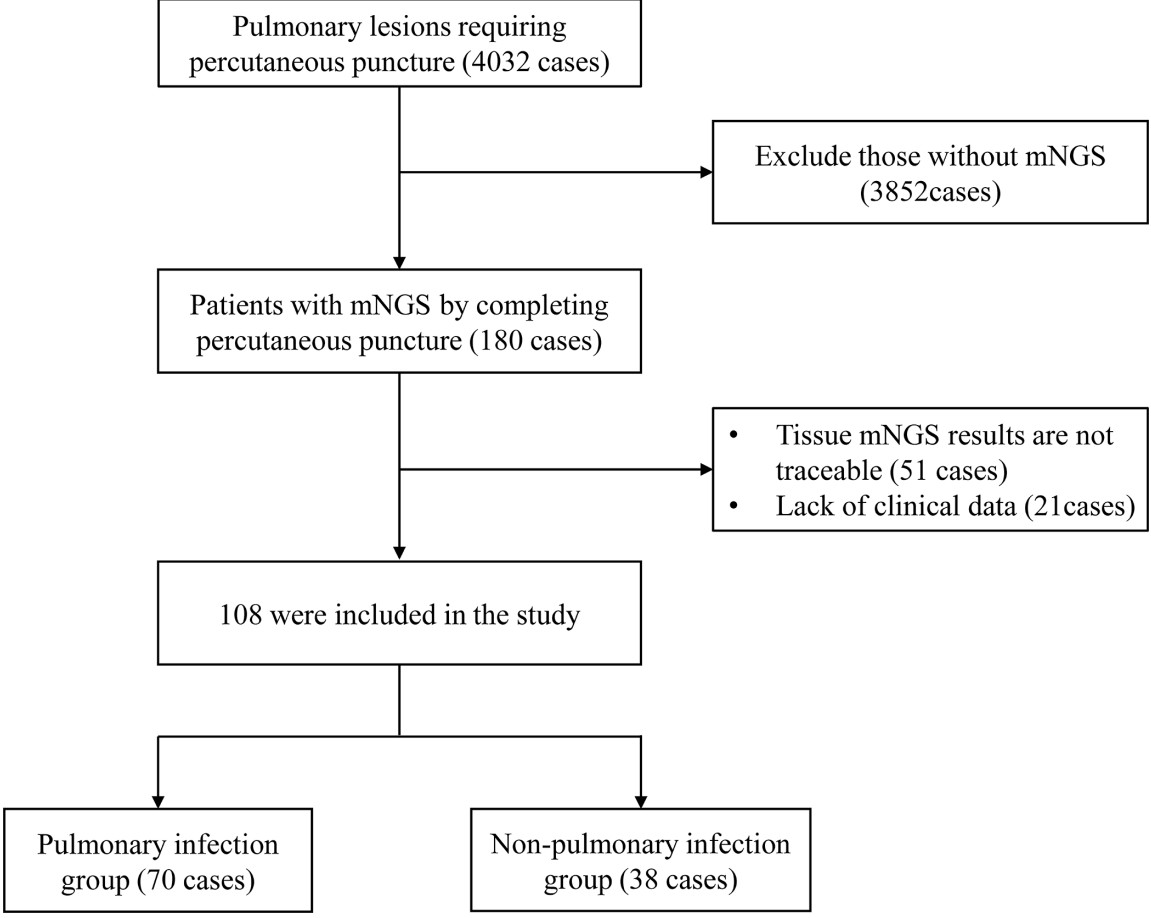

**FIG 1** Flowchart of patient screening.

then treated with lysozyme for 10 min. The BGI metagenomics library kit was used to prepare the library after lung tissue samples were split up by enzymatic digestion, end repair, primer adding, and PCR amplification. In this investigation, the sequencing depth was >20M reads using the MGIseq2000 sequencing technology with SE50. The Molecular Laboratory tests nucleic acids in lesion tissues to identify pathogenic microorganisms present in the samples, including 6,350 bacteria (including 133 *Mycobacteria* and 122 *Mycoplasma/Chlamydia/Rickettsia*), 1,798 DNA viruses, 1,064 fungi, and 234 parasites with known genetic sequences. Microorganisms were identified by comparing their nucleic acid sequences with those of microorganisms already in the database (using NCBI; ftp://ftp.ncbi.nlm.nih.gov/genomes). If the pathological section shows a non-neoplastic disease, the pathology department performs Gram staining, periodic acid Schiff staining (PAS staining), Gomori-Grocott methenamine silver stain (GMS), and Ziehl-Neelsen acid-fast bacilli staining (AFB) for bacterial, fungal, and tuberculosis detection. In addition, molecular nucleic acid testing will be added after the discovery of suspected pathogenic bacteria, such as TB-DNA and PCR for *Mycobacterium* spp., and PCR for *Aspergillus* spp. and *Nocardia* spp.

## Determination of results and diagnosis

The final diagnosis of 108 patients was determined by the patient's clinical presentation, imaging, histopathological findings, mNGS testing, and therapeutic effect. The final diagnosis is given by two separate clinicians based on the above information, and in the case of a controversial diagnosis, a third, more senior physician is asked to decide together.

Determination of histopathological findings: Positive results included the following: (i) tissue Gram staining suggests the presence of *cocci* or *bacilli*; (ii) special stains in pathology, such as periodic acid-Schiff stain (PAS stain), GMS, and AFB, are positive; (iii) pathological tissue sent for TBqPCR detects the presence of *Mycobacterium tuberculosis* DNA; (iv) the presence of certain pathogenic bacteria is clearly indicated in the pathology report; and (v) the pathology report suggests the presence of viral inclusion bodies. A negative result means that the above conditions are excluded and no clear pathogen is present.

Consider the presence of a corresponding pulmonary infection when mNGS has the following conditions: (i) consider the corresponding infection if *influenza virus*, *parainfluenza virus*, *respiratory syncytial virus*, *adenovirus*, etc., are detected in the patient's sample; (ii) consider the corresponding infection when microorganisms with difficulty breaking the wall are detected, such as *Mycobacterium bovis*, *Nocardia* spp., and *fungi*, regardless of their sequence number; (iii) identify rare pathogens, such as *Yersinia pestis* bacteria, dimorphic fungi (*Talaromyces marneffei*, *Histoplasma capsulatum*, *spores*, etc.), *Chlamydia psittaci*, and various parasitic infections, also considered infections; (iv) after excluding common colonizing microorganisms of human skin (*Propionibacterium acnes*, *Staphylococcus epidermidis*, etc.), if the pathogenic microbial sequences were greater than 100, they were considered infections; and (v) Aspiration pneumonia/lung abscess needs to be considered in patients at risk of aspiration or with a clear history of aspiration whose mNGS findings present with multiple common oral colonizing organisms.

## Statistical analysis

We used SPSS version 25.0 for statistical analysis. For the comparison of categorical variables between the two groups of patients, we used the Pearson chi-square test (Fisher's test if there was a theoretical number T < 1 or $n < 40$), while for the comparison of continuous variables, we used the *t* test of two independent samples. In addition, we plotted receiver operating characteristic curves (ROC curves) to compare the diagnostic efficacy of using histopathological tests alone with the combined use of histopathology and mNGS for infectious diseases. The statistical results of this paper are considered statistically significant when *P* is less than 0.05.

## RESULTS

### Patient characteristics

Among the 108 patients involved, 70 developed infectious diseases and 38 had noninfectious diseases. The demographic and clinical characteristics of the patients are listed in Table 1. There was no statistically significant difference between the infectious group and the noninfectious group according to age, sex, and course of the disease. The clinical diagnosis for these patients is shown in Table 2. In terms of imaging findings, 64 of these patients had predominantly solid imaging, and 5 patients had predominantly cavity formation on imaging.

### Laboratory test results

We retrospectively collected laboratory findings from 108 patients (Table 3). There was no significant difference in routine blood tests between the infectious disease group and the noninfectious disease group. Inflammatory indicators and the fungal 1–3-beta-D-glucan assay(G)/galactomannan (GM) test were also not significantly different between the two groups of patients. Among the 108 patients, the histopathological findings were significantly different between the infected and noninfected groups (*P* value = 0.004). Among the 108 patients, a total of 42 patients underwent tissue culture (including bacterial, fungal, and tuberculosis cultures), and 7 patients had positive culture results in the infected group (including *Mycobacterium tuberculosis*, *Penicillium* spp. *Nocardia*, *Cryptococcus*, *Klebsiella pneumoniae*, and *Enterobacter cloacae*) and no positive culture results in the noninfected group (*P* value = 0.052). In addition, the positive rate of mNGS

**TABLE 1** Demographic and clinical characteristics of the patients[a]

| Demographic and clinical characteristics | Total (*N* = 108) | Pulmonary infectious diseases (*N* = 70) | Pulmonary noninfectious diseases (*N* = 38) | *P* value |
|---|---|---|---|---|
| Age, mean (SD)—year | 51.08 (15.44) | 50.56 (14.98) | 52.05 (16.41) | 0.633 |
| Gender | | | | 0.075 |
| Male—no. (%) | 58 (53.7) | 42 (60.0) | 16 (42.1) | |
| Female—no. (%) | 50 (46.3) | 28 (40.0) | 22 (57.9) | |
| Course of disease, median (IQR)—day | 90 (30.0–292.5) | 105 (30.0–277.5) | 90 (30.0–365.0) | 0.416 |
| Symptom—no. (%) | | | | |
| Cough | 74 (68.5) | 49 (70.0) | 25 (65.8) | 0.653 |
| Expectoration | 57 (52.8) | 41 (58.6) | 16 (42.1) | 0.102 |
| Fever | 44 (40.7) | 29 (41.4) | 15 (39.5) | 0.843 |
| Chest pain | 30 (27.8) | 21 (30.0) | 9 (23.7) | 0.484 |
| Shortness of breath | 42 (38.9) | 25 (35.7) | 17 (44.7) | 0.358 |
| Hemoptysis | 23 (21.3) | 18 (25.7) | 5 (13.2) | 0.128 |
| Co-morbidity—no. (%) | | | | |
| Diabetes | 9 (8.3) | 8 (11.4) | 1 (2.6) | 0.114 |
| Hypertension | 20 (18.5) | 15 (21.4) | 5 (13.2) | 0.291 |
| Immunosuppressive factors | 19 (17.6) | 12 (17.1) | 7 (18.4) | 0.868 |
| Chronic lung disease | 12 (11.3) | 7 (10.3) | 5 (13.2) | 0.655 |
| COPD | 2 | 1 | 1 | |
| Emphysema | 3 | 0 | 3 | |
| Bronchiectasis | 6 | 3 | 3 | |
| Others | 3 | 3 | 0 | |

[a]SD, standard deviation; no, number; IQR, interquartile range; COPD, chronic obstructive pulmonary disease.

in the infected group was significantly higher than that in the noninfected group (*P* value < 0.01).

## Comparison of two testing protocols

We compared the diagnostic efficiency of histopathological testing plus molecular nucleic acid testing (PCR) and histopathological testing combined with tissue mNGS in diagnosing infectious diseases. Pathogens detected by histopathology testing plus molecular nucleic acid testing (PCR) were *Mycobacterium* spp. (10 patients), *Actinomyces* spp. (1 patient), and *Aspergillus* spp. (1 patient), *Cryptococcus* spp. (1 patient), *Filamentous bacteria* (1 patient), and yeast-like fungi (2 patients). The main pathogenic bacteria detected in the included patients using tissue mNGS were *Mycobacterium* spp., *Porphyromonas* spp., *Cryptococcus* spp., *Micrococcus* spp., *Prevotella* spp., *Treponema* spp., *Streptococcus* spp., *Actinobacteria* spp., *Campylobacter* spp., *Fusobacterium* spp., *Aspergillus* spp., etc. Table S1 shows the major pathogen lineages in 108 patients. It can be seen from Table S1 that the types of pathogenic bacteria detected by the combined organization of mNGS are far greater than those detected by traditional histopathological testing alone. To compare the diagnostic efficacy of traditional histopathological testing and tissue mNGS combined with histopathological testing for infectious diseases, we plotted the ROC curves of the two testing protocols. Figure 2 shows the ROC curve for both testing protocols, with an area under the curve of 0.840 for infectious disease detection using mNGS in combination with histopathology testing (the area under the ROC curve when using histopathological methods alone is 0.644). For pulmonary fungal infections, the area under the ROC curve for the combined use of mNGS with histopathology was 0.876, compared with 0.760 for histopathology testing alone. For the diagnosis of *Mycobacterium tuberculosis*, the area under the ROC curve was 0.875 for the combined use of mNGS and 0.719 for the histopathological assay alone. The ROC curve for the diagnosis of lung abscess using the combination of mNGS was 0.840 compared with 0.575 for histopathology alone.

**TABLE 2** Clinical diagnosis of enrolled patients[a]

|  | No. (%) |
| --- | --- |
| Pulmonary infectious diseases | **70 (64.8)** |
| Mycobacterium infection | 18 (16.7) |
| Pulmonary tuberculosis | 16 (14.8) |
| Lung abscess | 20 (18.5) |
| Pneumonia | 10 (9.3) |
| Fungal infection | 14 (13.0) |
| Cryptococcal pneumonia | 7 (6.5) |
| Pulmonary aspergillosis | 4 (3.7) |
| Pneumocystis jirovec pneumonia | 1 (0.9) |
| Penicillium marneffei pneumonia | 1 (0.9) |
| Other infectious diseases | 8 (7.4) |
| Non-neoplastic and noninfectious diseases |  |
| Organizing pneumonia | 9 (8.3) |
| Non-specific interstitial pneumonia | 2 (1.9) |
| IPAF | 1 (0.9) |
| ANCA-associated vasculitis | 2 (1.9) |
| Rheumatoid arthritis | 1 (0.9) |
| Other noninfectious diseases | 8 (2.0) |
| Oncological diseases |  |
| Lung cancer | 13 (12.0) |
| Lymphoma | 3 (2.8) |

[a]ANCA, antineutrophil cytoplasmic antibodies; IPAF, interstitial pneumonia with autoimmune features.

## Case study

Of the 20 patients with a final diagnosis of lung abscess, five had histopathological findings that suggested that they had fungal infection, and we performed a detailed case study of two representative cases of these patients. In both cases, the pathologists misidentified the pathogen as a fungus, and antifungal treatment was ineffective. The resorption of the lesion after antibacterial treatment suggested the absence of fungal infection and confirmed the diagnosis of mNGS.

The first patient was a 70-year-old man who was admitted with recurrent cough and hemoptysis for more than eight years and had a history of dental caries. His laboratory test results suggested a slightly higher white blood cell count of $10.28 \times 10^9$/L and

**TABLE 3** Laboratory test results of enrolled patients[a]

| Laboratory test results | Total (N = 108) | Pulmonary infectious diseases (N = 70) | Pulmonary noninfectious diseases (N = 38) | P value |
| --- | --- | --- | --- | --- |
| WBC ($\times 10^9$/L)—median (IQR) | 6.89 (5.29–9.81) | 6.98 (5.38–9.99) | 6.51 (5.20–8.81) | 0.440 |
| N ($\times 10^9$/L)—mean (SD) | 5.71 (3.74) | 5.91 (3.96) | 5.36 (3.34) | 0.473 |
| L ($\times 10^9$/L)—mean (SD) | 1.50 (0.72) | 1.54 (0.78) | 1.40 (0.60) | 0.334 |
| Hb (g/L)—mean (SD) | 116.94 (25.24) | 119.71 (26.30) | 111.92 (22.67) | 0.127 |
| PCT (ng/mL)—median (IQR) | 0.05 (0.02–0.16) | 0.05 (0.02–0.20) | 0.05 (0.02–0.09) | 0.332 |
| CRP (mg/mL)—median (IQR) | 17.00 (3.00–88.30) | 18.10 (3.55–109.00) | 13.40 (2.57–55.30) | 0.212 |
| ESR (mm/h)—mean (SD) | 51.49 (39.28) | 52.02 (39.52) | 50.36 (39.54) | 0.863 |
| G test positive—no. (%) | 4 (4.2) | 3 (4.8) | 1 (2.9) | 0.656 |
| GM test positive—no. (%) | 2 (2.0) | 2 (3.0) | 0 (0.0) | 0.305 |
| Traditional histopathological detection methods—no. (%) | 22 (20.6) | 20 (29.0) | 2 (5.3) | 0.004 |
| Lung tissue culture positive— no. (%) | 7 (16.7) | 7 (24.1) | 0 (0.0) | 0.052 |
| mNGS positive—no. (%) | 57 (52.8) | 54 (77.1) | 3 (7.9) | 0.000 |

[a]IQR, interquartile spacing; WBC, leukocyte count; N, neutrophil count; L, lymphocyte count; Hb, hemoglobin; PCT, calcitoninogen; CRP, C-reactive protein; ESR, erythrocyte sedimentation rate; G test, fungal 1-3-beta-D-glucan assay; GM test, galactomannan test.

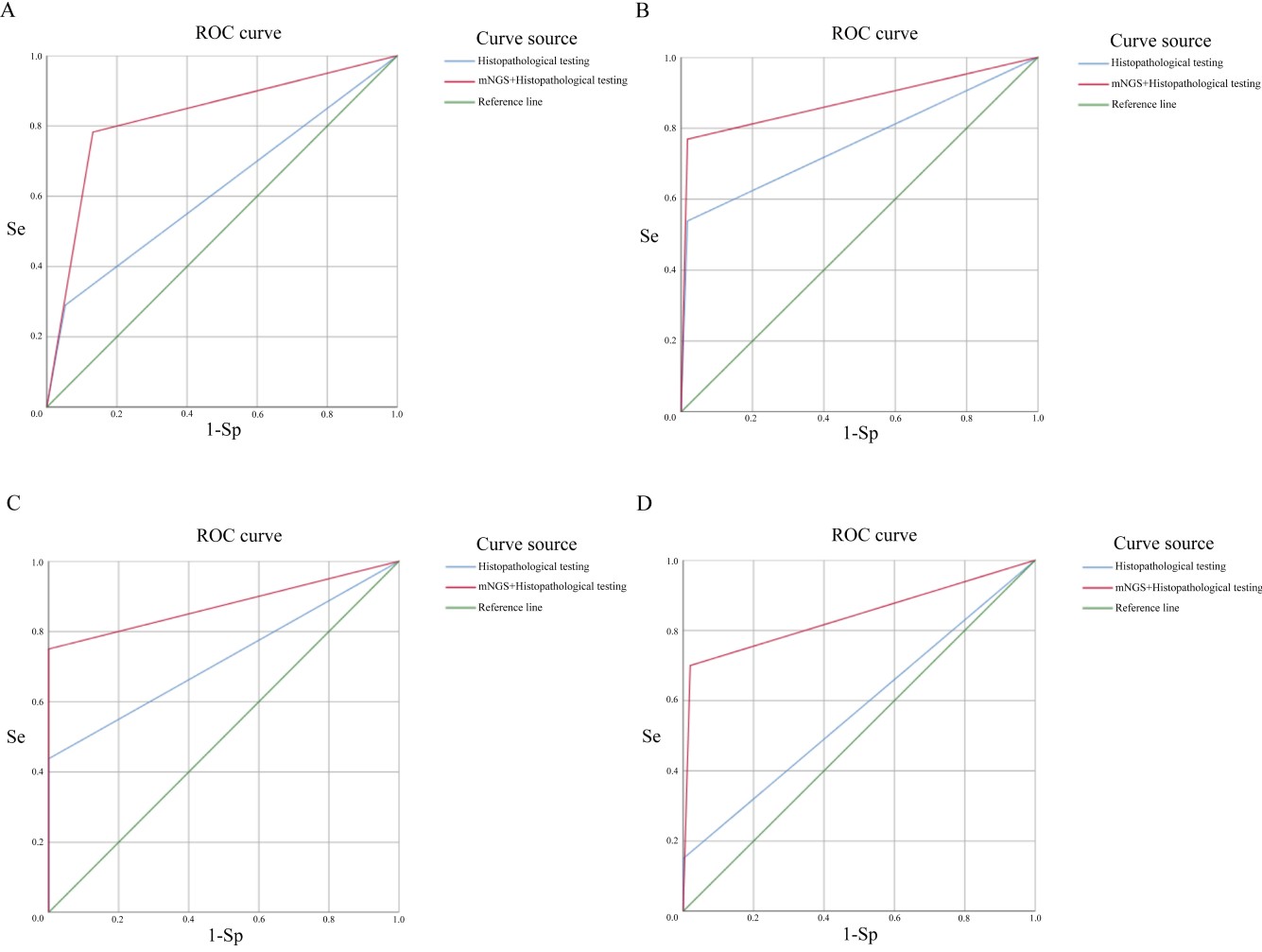

**FIG 2** ROC curves for both testing assays. The red line represents combined mNGS and histopathology testing, and the blue line represents histopathology testing used alone. (A) Diagnostic efficacy in pulmonary infectious diseases. (B) Diagnostic efficacy of pulmonary fungal infection. (C) Diagnostic efficacy of pulmonary tuberculosis. (D) Diagnostic efficacy of lung abscess.

an ESR greater than 120 mm/h. His chest CT scan showed a large solid lesion in the left lower lung with a small amount of exudative shadow around the lesion (Fig. 3A and B). Histopathological findings suggested chronic inflammation of the mucosa with granulomatous hyperplasia (Fig. 3C), and a small amount of mycelium was detected with positive GMS (Fig. 3D). The patient's symptoms did not improve after antifungal treatment with fluconazole. After the mNGS results were shown, it was suggested that there was a mixed infection of various bacteria, such as *Porphyromonas* spp., *Clostridium* spp., *Tannerella forsythia*, and *Spirochetes* spp. After administered with piperacillin sodium tazobactam sodium, the patient's cough and hemoptysis improved compared to before.

The second patient was a 35-year-old woman who was admitted with recurrent cough and sputum with hemoptysis for more than two years. She was previously diagnosed with acute myeloid leukemia, type M4, and underwent allogeneic hematopoietic stem cell transplantation. Her CT scan of the chest suggested solid changes in the right lower lung with cavity formation, and the cavity contained spherical hyperdense shadows (Fig. 4A and B). Her histopathological findings tended to be mucormycosis, with positive results for PAS and GMS (Fig. 4C and D). There was no improvement after amphotericin B anti-infection treatment. However, the mNGS results suggested a mixed infection of *Streptococcus* spp., *Parvimonas* spp., *Prevotella* spp., *Campylobacter*

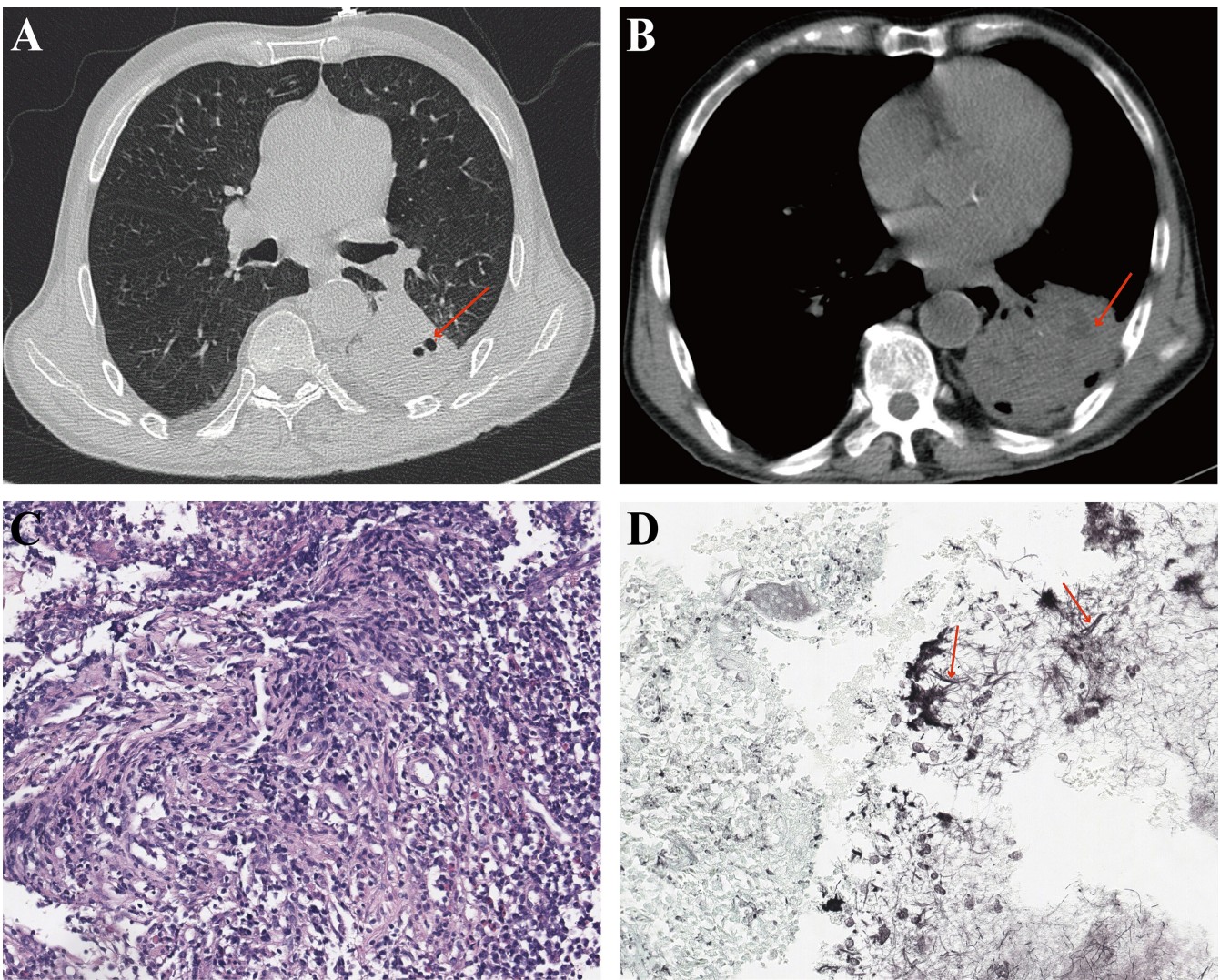

**FIG 3** Chest HRCT and pathological manifestations of the first patient. (A) The pulmonary window of chest HRCT showed a large solid shadow in the left lower lung with small cavity formation. (B) The mediastinal window of chest HRCT showed a small amount of liquefaction formation in the left lower lung. (C) Hematoxylin–eosin stain of lung biopsy tissue showed chronic inflammatory exudation. (D) GMS of the lesioned tissue showed filamentous microorganisms which were misdiagnosed as fungal hyphae (red arrows).

spp., *Clostridium* spp., *Porphyromonas* spp., *Haemophilus* spp., and *Prevotella* spp. The patient's symptoms improved after receiving anti-infection treatment with piperacillin tazobactam.

## DISCUSSION

mNGS is now widely used clinically for pathogenic testing of infectious diseases, such as hematologic infections (8), central nervous system infections (9, 10), and respiratory infections (11). A growing number of studies have confirmed that mNGS has higher sensitivity and specificity than traditional pathogenic assays in the diagnosis of infectious diseases (3, 6). In studies related to pulmonary infectious diseases, alveolar lavage fluid is used for most specimens (12, 13), which has the advantage of easy access over percutaneous lung puncture biopsy. However, in general, BALF specimens have the following problems: first, it has been confirmed that the human lung environment is nonsterile (14, 15), and alveolar lavage fluid can partially represent the human airway microbial situation, but whether it can adequately represent the microbial situation in the diseased

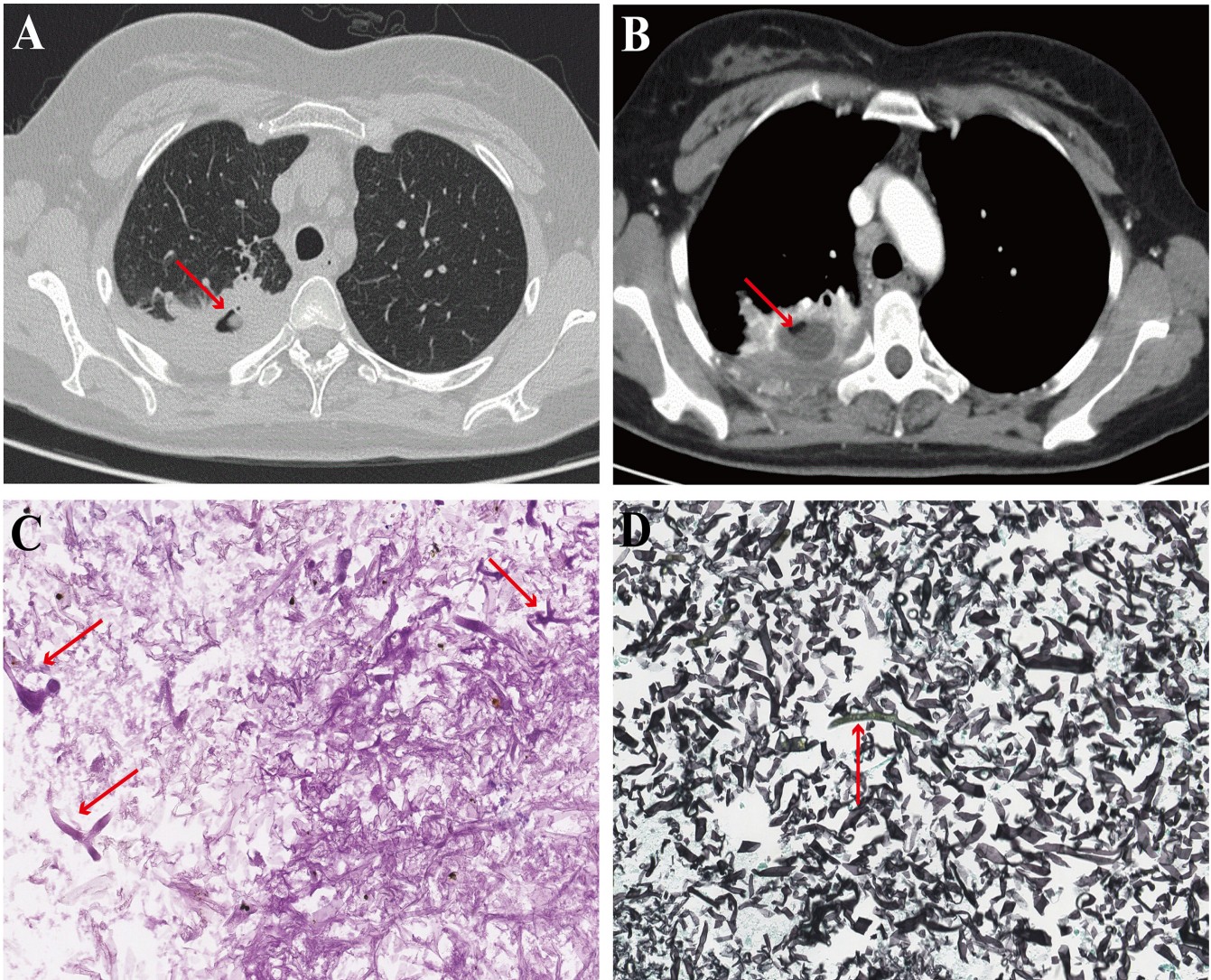

**FIG 4** Chest HRCT and pathological manifestations of the second patient. (A) The pulmonary window of chest HRCT showed a large *dimorphic fungi* solid shadow with cavity formation in the lower lobe of the right lung. (B) The enhanced mediastinal window of chest HRCT showed a circular enhanced lesion around the cavity. (C) PAS stain of lung biopsy tissue showed coarse hyphae forming a right angle (red arrows). (D) GMS of the lesioned tissue showed coarse hyphae forming a right angle, which was misdiagnosed as mucor (red arrows).

lung has not been confirmed (16); second, when performing BALF, the operator only lavages the approximate location of the lesion, and there is no way to clarify whether the lavage fluid adequately contacts the location of the lesion, so whether the lavage fluid can adequately elute microorganisms from the lesion site cannot be determined. In addition, the fiberoptic bronchoscope must pass through the upper respiratory tract, and pollution from the upper respiratory tract cannot be excluded. Therefore, we chose a more intuitive sampling method, CT-guided percutaneous lung aspiration biopsy, in which the lesion tissue taken under direct CT view is subjected to histopathology and mNGS testing. Similar studies have been performed previously, but the sample size was too small (4). The tissue we sent for examination was part of the diseased lung tissue obtained under real-time CT guidance; therefore, it was well representative of the microorganisms at the lesion site.

In this study, we compared the diagnostic efficacy of histopathological testing alone with the combined use of mNGS and histopathological testing for various infectious diseases. Combining mNGS with histopathological methods was able to detect more

*Mycobacterium* spp., *Cryptococcus* spp., *Actinomyces* spp., *Aspergillus* spp., and *Streptococcus* spp. than histopathological testing alone, and in addition, the combined testing protocol improved the detection rate of some rare pathogens, such as *Talaromyce* spp. and *Nocardia* spp. These results show that combining mNGS with traditional pathogen detection methods can greatly improve the positive rate of detection results. In addition, mNGS can compensate for the low detection rate of some rare pathogens with high culture requirements. We know that a lung abscess is a mixed infection. The positive rate of pathological special staining is low, and the morphology of pathogenic bacteria is not specific, which can be easily misdiagnosed as other pathogens; the positive rate of microbial culture is also low, especially for anaerobic infections; in our cases, the culture positivity rate was only 16.7%. Besides, the culture time is long; in our cases, it takes 24 hours for mNGS to get results, 7–10 days for histopathology section preparation and special staining, and about 3 days for common bacteria, 5–7 days for *Nocardia*, and 21 days for *Mycobacterium* in tissue culture, and only the dominant flora can be cultured. Unlike the previous two testing methods, mNGS can detect anaerobic bacteria and specific pathogens and find the full range of pathogens in mixed infections. Related literature also reports that mNGS also detects *Orientia tsutsugamushi*, *Nocardia otitidiscaviarum*, *Chlamydia psittaci*, and other rare pathogens (17–19). Moreover, it has been reported in the literature that mNGS is less likely to be affected by the use of antibiotics before obtaining the sample than traditional pathogenic tests (3). For the detection of tuberculosis, the combined testing protocol improved the diagnostic rate of *Mycobacterium tuberculosis*. Therefore, combining mNGS with histopathological testing is an ideal option to assist in the clinical diagnosis of infectious diseases.

Liu et al. (16) compared the diagnostic efficacy of mNGS for infectious diseases using transbronchial lung biopsy (TBLB) tissue and BALF and found that TBLB tissue and BALF have respective advantages in specificity and sensitivity for mNGS analysis. Wang et al. used mNGS analysis to compare specimens from 39 patients with TBLB, BALF, and bronchial needle brush (BB) and found that TBLB had higher specificity, but there was no statistically significant difference in the sensitivity of the three assays (20). We found that the combined use of mNGS not only significantly improved the diagnostic efficacy of infectious diseases but also well distinguished the pathogenic microorganism of infectious diseases by using lung puncture biopsy tissue. However, as far as the ROC curve is concerned, the sensitivity and specificity of combined mNGS with conventional histopathological testing by using biopsy tissue are high, which seems to improve the diagnostic efficacy compared with the sensitivity and specificity mentioned in previous studies using alveolar lavage fluid (7, 21). Therefore, whether the tissue obtained by lung puncture is superior to that obtained by TBLB needs further study.

## Conclusion

In conclusion, in this study, we performed both mNGS and pathology on lung puncture biopsy tissue from patients and found that combined mNGS and histopathology testing was significantly more effective than histopathology testing alone in detecting infectious diseases and identifying infectious diseases. In addition, the combined approach improves the detection rate of pathogenic microorganisms in infectious diseases and can be used to guide precision clinical treatment.

## ACKNOWLEDGMENTS

The authors would like to thank the pathology staff, such as Weiya Wang, and the respiratory staff for their help and support of our study.

This work was supported by the National Natural Science Foundation of China (grant number 82173182) and the Science and Technology Program of Sichuan, China (grant number 2020YFS0572).

All the authors contributed substantially to the work presented in this article. D.L., W.M.L., K.G.W., and L.H.Y. conceived the study. K.G.W. and L.H.Y. contributed to data

interpretation. Y.L.L. is mainly responsible for specimen collection and processing. K.G.W. and L.H.Y. contributed to the study protocol and wrote the article. Y.L.L., D.L., and W.M.L. revised the article. The corresponding author had full access to all of the data and the final responsibility for the decision to submit this article for publication.

## AUTHOR AFFILIATION

[1]Department of Pulmonary and Critical Care Medicine, West China Hospital, Sichuan University, Chengdu, China

## AUTHOR ORCIDs

Linhui Yang  http://orcid.org/0009-0007-9669-3380
Weimin Li  http://orcid.org/0000-0003-0985-0311
Dan Liu  http://orcid.org/0000-0002-6618-0150

## FUNDING

| Funder | Grant(s) | Author(s) |
| --- | --- | --- |
| MOST | National Natural Science Foundation of China (NSFC) | 82173182 | Dan Liu |
| Science and Technology Program of Sichuan, China | 2020YFS0572 | Yalun Li |

## AUTHOR CONTRIBUTIONS

Linhui Yang, Data curation, Formal analysis, Software, Visualization, Writing – original draft | Kaige Wang, Data curation, Investigation, Project administration, Writing – review and editing | Yalun Li, Resources, Writing – review and editing | Weimin Li, Funding acquisition, Writing – review and editing | Dan Liu, Funding acquisition, Project administration, Supervision, Writing – review and editing

## DATA AVAILABILITY

As our study was retrospective, the mNGS results of our patients were obtained through the hospital's medical record system. Raw data obtained from the Center for Precision Medicine at West China Hospital are available via FigShare. The datasets used and/or analyzed in the present study are available from the corresponding author on reasonable request.

## ETHICS APPROVAL

This study has been approved by the Biomedical Ethics Review Committee of West China Hospital of Sichuan University with the ethics number "2022 Review 1628." The committee agreed to waive the informed consent form.

## ADDITIONAL FILES

The following material is available online.

### Supplemental Material

**Table S1 (Spectrum00586-23-S0001.xlsx).** Results of pathogenic microorganisms detected by histopathology and mNGS in 108 patients.

### Open Peer Review

**PEER REVIEW HISTORY (review-history.pdf).** An accounting of the reviewer comments and feedback.

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
