## [Reviewer comments · Microbiology Spectrum]

Microbiology Spectrum

The joint application of metagenomic next-generation sequencing and histopathological examination for the diagnosis of pulmonary infectious disease

Linhui Yang, Kaige Wang, Yalun Li, Dan Liu, and Weimin Li

Corresponding Author(s): Dan Liu, Sichuan University West China Hospital

Review Timeline:

Submission Date:	February 14, 2023
Editorial Decision:	June 20, 2023
Revision Received:	July 14, 2023
Accepted:	August 28, 2023

Editor: Arryn Craney

Reviewer(s): The reviewers have opted to remain anonymous.

Transaction Report:

DOI: <https://doi.org/10.1128/spectrum.00586-23>

June 20, 2023

Prof. Dan Liu
Sichuan University West China Hospital
No.37 Guoxue Lane
Chengdu,Sichuan,China
China

Re: Spectrum00586-23 (The joint application of metagenomic next-generation sequencing and histopathological examination for the diagnosis of pulmonary infectious disease)

Dear Prof. Dan Liu:

While this work is timely and will benefit those in the field, both reviewers recommended extensive modifications to the manuscript. Of note, clarity in the clinical impact was mentioned by both reviewers and the section "case study" should be renamed and delve into clinical impact of combined NGS + histopathological vs standard of care. I look forward to reading your revised manuscript submission. Please address the reviewer comments below.

Link Not Available

Sincerely,

Arryn Craney

Journals Department
Reviewer comments:

Reviewer #2 (Comments for the Author):

The authors' manuscript "The joint application of metagenomic next-generation sequencing and histopathological examination for the diagnosis of pulmonary infectious disease" describes the combination of molecular sequencing and histopathology to diagnose infections in the pulmonary setting. The authors conclude that metagenomic NGS testing is superior to histopathologic examination and that combined metagenomic testing and histopathologic examination has the highest degree of sensitivity. The

results are timely and of interest, but the presentation of the result data does not provide clear insight into the clinical significance of the increased sensitivity provided by mNGS. I have specific suggestions for improvement.

Major suggestions:

1. Line 163; while the morphology of microorganisms on histopathologic examination affords some suggestion of genus, in my experience it is not typically possible to definitively identify the genus of a microorganism in a sample (for example, filamentous bacteria could potentially represent actinomyces or nocardia, while hyaline fungal hyphae could represent aspergillum or fusarium or many other fungi); correlative culture or molecular testing may resolve the issue, but in our practice we would not describe microorganisms at the genus level in most cases by morphology alone. It might be more appropriate to indicate the the general morphologies that were observed by histopathology. Extending from this thought, the data presented in table 2 illustrates the incremental improvement in genus level identification of microorganisms utilizing mNGS. There is little surprise (in my opinion) that additional organisms would be identified at the genus level; I am not sure detailing each organism in this way clarifies the value of the additional diagnostic sensitivity. I this this figure (or a simple table of these organisms) could be added to appendix material rather than presented here.
2. Figure 3. I'm not certain that the performance differences between histopathologic examination alone, mNGS alone and the combination is best represented in this manner. I would suggest instead presenting the data in a simple tabular format wherein the performance differences for bacteria / fungi / mycobacteria by case are more easily appreciated. For instance, I cannot tell from this analysis if there were any cases where histopathology was successful but mNGS was not. Similarly, is the increased "sensitivity" of mNGS in this context a factor of detecting additional organisms per case or organisms in additional cases of both?
3. It would be helpful if the clinical significance of the additional sensitivity reported for mNGS were more clearly illustrated. The two provided case examples show lesions that were histopathologically identified as fungal infections and yet were identified as bacterial infections by mNGS. Did mNGS also identify the fungal infections? Is there a rationale for why a polymicrobial bacterial infection (one assumes abscess) was not detected by histopathologic examination? Is sampling (different cores sent for pathology and molecular testing) a potential reason?
4. The low-sensitivity and poor turnaround time of culture is referenced in the paper, but comparative turnaround time for mNGS and histopathologic examination is not provided - it should be.
5. Additional details on the mNGS method should be provided. How was the nucleic acid extracted and how were libraries prepared? What sequencing platform was employed? What was the average depth of sequencing? If this has been published elsewhere, the method could be referenced.

Minor suggestions:

1. Line 30; the phrase "punctured lesion tissue" may not be familiar to all readers, I might suggest "core needle biopsy tissue" or some other surrogate descriptor.
2. Line 55; it may be helpful to quantify the positive rate of culture (described as 'low') and provide a reference to help contextualize the results for the reader
3. Line 79; The construction of the sentence beginning "The preliminary clinical diagnosis..." Is somewhat awkward, I would suggest revising the sentence to make the key points more clear.
4. Line 93; A 'lyophilized tube containing 10% formalin' is referenced; this may simply be out of my experience, but I have not encountered 10% formalin in a non-liquid preparation (in our practice is typically buffered formalin in water). Is the descriptor lyophilized appropriate here?
5. Line 104; the stain descriptors "hexamine silver staining, and antacid staining" are less familiar to me and may not be readily understood by all readers. Would these stains be equivalent to Gomori-Grocott methenamine silver stain (GMS) and/or Ziehl-Neelsen acid-fast bacilli staining (AFB)?
6. Microorganism genus and species names should be italicized
7. Line 125; the descriptor "bidirectional fungi" may be less familiar than "dimorphic fungi"
8. Line 125; "Penicillium marneffeii" was renamed "Talaromyces marneffeii" in 2015; it may be helpful to employ the updated name.
9. Line 151; the expanded form of G/GM tests should be written before the abbreviations are employed
10. Line 159; I assume the P-value does not actually equal 0.000, perhaps a less than symbol be employed?
11. Line 197; was a beta-lactamase inhibitor used in combination with penicillin as stated, or another beta-lactam antibiotic (such as ampicillin)?

Reviewer #3 (Comments for the Author):

Major issues:

- Culture is often not ordered when obtaining a lung biopsy and an infectious process is only suspected after pathology has been performed - comparison of the culture data with the mNGS results would be useful to assess performance. Culture remains the diagnostic standard, from the patients enrolled, how many specimens were actually processed for culture?
- This manuscript is either proposing mNGS as a diagnostic tool or a case report - see Case studies section - consider publishing those separately. This manuscript could be more focused
- Pathogen abundance in the mNGS data is missing, how significance was assessed in the case of polymicrobial findings
- mNGS detailed procedure or reference to a detailed procedure is missing

Minor issues:

- Line 88 - disease progression? Instead of regression
- Scientific names need to be reviewed
 - o Line 125 *Penicillium marneffeii* is now known as *Talaromyces marneffeii*
 - o Line 125 ?*Podocystis* - do not know of a clinically relevant fungi named *Podocystis histolytica* - please review and correct
 - o Line 158 *Klebsiella pneumoniae*
 - o Line 197 - what is *Clostridium forcestorum* spp??
 - o Line 208 - what is *Microsomonas*
- Line 146 - imaging findings? Instead of performance
- Review paragraph line 163 - special stains in histopathology can direct clinician to potential diagnostics but not detect pathogens: pathology would find acid fast bacilli resembling *Mycobacterium* or fungal elements/hyphae resembling *Aspergillus*, but cannot name the pathogens as implied on this statement
- Line 174 - abbreviations should be defined first time used - ROC
- Line 238 - *Penicillium* spp. is usually a contaminant not a rare pathogen

Staff Comments:

Preparing Revision Guidelines

Please return the manuscript within 60 days; if you cannot complete the modification within this time period, please contact me. If you do not wish to modify the manuscript and prefer to submit it to another journal, please notify me of your decision immediately so that the manuscript may be formally withdrawn from consideration by Microbiology Spectrum.

Review manuscript: The joint application of metagenomic next-generation sequencing and histopathological examination for the diagnosis of pulmonary infectious disease

Authors: Yang L, et al

Summary:

The authors present the comparison between mNGS and traditional pathological examination for the diagnosis of pulmonary infectious diseases. Metagenomics has been proven to be a powerful tool for the diagnosis of occult infections where other techniques may have limitations in sensitivity. Lung infectious are particularly difficult to diagnose for reasons the authors stated and often also because an infectious process is not suspected until after pathology has been performed.

Major issues:

- Culture is often not ordered when obtaining a lung biopsy and an infectious process is only suspected after pathology has been performed – comparison of the culture data with the mNGS results would be useful to assess performance. Culture remains the diagnostic standard, from the patients enrolled, how many specimens were actually processed for culture?
- This manuscript is either proposing mNGS as a diagnostic tool or a case report – see Case studies section - consider publishing those separately. This manuscript could be more focused
- Pathogen abundance in the mNGS data is missing, how significance was assessed in the case of polymicrobial findings
- mNGS detailed procedure or reference to a detailed procedure is missing

Minor issues:

- Line 88 – disease progression? Instead of regression
- Scientific names need to be reviewed
 - o Line 125 *Penicillium marneffeii* is now known as *Talaromyces marneffeii*
 - o Line 125 ?*Podocystis* – do not know of a clinically relevant fungi named *Podocystis histolytica* - please review and correct
 - o Line 158 *Klebsiella pneumoniae*
 - o Line 197 – what is *Clostridium forcestorum* spp??
 - o Line 208 – what is *Microsomonas*
- Line 146 – imaging findings? Instead of performance
- Review paragraph line 163 – special stains in histopathology can direct clinician to potential diagnostics but not detect pathogens: pathology would find acid fast bacilli resembling *Mycobacterium* or fungal elements/hyphae resembling *Aspergillus*, but cannot name the pathogens as implied on this statement
- Line 174 – abbreviations should be defined first time used – ROC
- Line 238 – *Penicillium* spp. is usually a contaminant not a rare pathogen

Responses to Reviewers

July 8, 2023

Arryn Craney

Microbiology Spectrum

Dear editor,

Thank you for considering our revised manuscript "**The joint application of metagenomic next-generation sequencing and histopathological examination for the diagnosis of pulmonary infectious disease**" (Manuscript ID: Spectrum00586-23) for publication in Microbiology Spectrum. We appreciate the constructive comments/suggestions provided by the reviewers. We have revised this manuscript to address the concerns raised by the reviewers, and all changes in the revision are highlighted in red font. Our point-by-point replies are listed below. We hope that the revised version of the manuscript is now suitable for publication in *Microbiology Spectrum*. We are willing to provide further information if necessary.

None of the authors have any conflicts of interests or financial disclosures to declare.

Thank you again for considering our revised manuscript! If you have any queries, please don't hesitate to contact me at the address by liudan10965@wchscu.cn.

Sincerely,

Dan Liu

Reviewer comments:

Reviewer #2:

The authors' manuscript "The joint application of metagenomic next-generation sequencing and histopathological examination for the diagnosis of pulmonary infectious disease" describes the combination of molecular sequencing and histopathology to diagnose infections in the pulmonary setting. The authors conclude that metagenomic NGS testing is superior to histopathologic examination and that combined metagenomic testing and histopathologic examination has the highest degree of sensitivity. The results are timely and of interest, but the presentation of the result data does not provide clear insight into the clinical significance of the increased sensitivity provided by mNGS. I have specific suggestions for improvement.

Answer: Thank you for your suggestions and recommendations.

Major suggestions:

1. Line 163; while the morphology of microorganisms on histopathologic examination affords some suggestion of genus, in my experience it is not typically possible to definitively identify the genus of a microorganism in a sample (for example, filamentous bacteria could potentially represent actinomyces or nocardia, while hyaline fungal hyphae could represent aspergillum or fusarium or many other fungi); correlative culture or molecular testing may resolve the issue, but in our practice we would not describe microorganisms at the genus level in most cases by morphology alone. It might be more appropriate to indicate the the general morphologies that were observed by histopathology. Extending from this thought, the data presented in table 2 illustrates the incremental improvement in genus level identification of microorganisms utilizing mNGS. There is little surprise (in my opinion) that additional organisms would be identified at the genus level; I am not sure detailing each organism in this way clarifies the value of the additional diagnostic sensitivity. I this this figure (or a simple table of these organisms) could be added to appendix material rather than presented here.

Answer: Thank you for your suggestions and recommendations. As suggested, molecular nucleic acid testing will be added after finding pathogenic bacteria in pathological tissue, such as TB-DNA for Mycobacterium, and molecular PCR for Aspergillus and Nocardia to identify pathogens. Therefore, "histopathological testing alone" may confuse the reader, and we changed "histopathological testing alone" to "pathological tissue + molecular nucleic acid testing (PCR)" (**line 181**). In addition, we modified the means of histopathological examination into the sentence "In addition, molecular nucleic acid testing will be added after the discovery of suspected pathogenic bacteria, such as TB-DNA and polymerase chain reaction (PCR) for Mycobacterium spp., and PCR for Aspergillus spp. and Nocardia spp." (**line 120**). Besides, we have removed Figure 2 in response to your comments and replaced it with a more informative **Table S1**. Thanks for your professional consideration and suggestions, and we believe the revised manuscript would be more improved according to your suggestion.

2. Figure 3. I'm not certain that the performance differences between histopathologic examination alone, mNGS alone and the combination is best represented in this manner. I would suggest instead presenting the data in a simple tabular format wherein the performance differences for bacteria / fungi / mycobacteria by case are more easily appreciated. For instance, I cannot tell from this analysis if there were any cases where histopathology was successful but mNGS was not. Similarly, is the increased "sensitivity" of mNGS in this context a factor of detecting additional organisms per case or organisms in additional cases of both?

Answer: Thank you for your suggestions and recommendations. After careful consideration, we thought that the ROC curve can be used to compare the diagnostic efficacy of the two protocols. The main roles of ROC curves are (1) making it easy to find out the ability of a classifier to recognize a sample at a certain threshold value. (2) selecting the best diagnostic threshold value of a diagnostic method. [2] The closer the ROC curve is to the upper left corner, the higher the sensitivity and the lower the false positive rate, the better the performance of the diagnostic method. (3) comparing the ability of two or more different diagnostic methods to identify a disease [2]. When comparing two or more diagnostic

methods for the same disease, the ROC curves of each diagnostic method can be drawn in the same ROC space, so that the advantages and disadvantages of each diagnostic method can be visually identified. The ROC curve near the top left corner represents the better performance of the diagnostic method. Besides, the increased sensitivity of mNGS is due to the ability to detect more pathogenic microorganisms in each case. In addition, the test results for each patient have been added in **Table S1**. Thanks for your consideration and suggestions, and we believe the revised manuscript would be more improved according to your suggestion.

Reference:

1. Dorin RI, Qualls CR, Crapo LM. Diagnosis of adrenal insufficiency. *Annals of internal medicine* 2003;139:194-204.
2. Mandrekar JN. Receiver operating characteristic curve in diagnostic test assessment. *Journal of thoracic oncology : official publication of the International Association for the Study of Lung Cancer* 2010;5:1315-6.

3. It would be helpful if the clinical significance of the additional sensitivity reported for mNGS were more clearly illustrated. The two provided case examples show lesions that were histopathologically identified as fungal infections and yet were identified as bacterial infections by mNGS. Did mNGS also identify the fungal infections? Is there a rationale for why a polymicrobial bacterial infection (one assumes abscess) was not detected by histopathologic examination? Is sampling (different cores sent for pathology and molecular testing) a potential reason?

Answer: Thank you for your suggestions and recommendations. Both methods used core needle biopsy tissue (which we have described in detail in **Line 102**). As suggested, we carefully compared the histopathological findings with the mNGS findings in both patients and found that mNGS detected the presence of a mixture of microorganisms (**Table S1**). While diagnosis of pathogenic bacteria by morphology has some errors. In both cases, the pathologists misidentified the pathogen as a fungus and antifungal treatment was ineffective. Resorption of the lesion after antibacterial treatment suggested the absence of fungal infection and confirmed the diagnosis of mNGS. Therefore, we explain it accordingly at **line 206**. Thanks for your consideration and suggestions, and we believe the revised manuscript would

be more improved according to your suggestion.

4. The low-sensitivity and poor turnaround time of culture is referenced in the paper, but comparative turnaround time for mNGS and histopathologic examination is not provided - it should be.

Answer: Thank you for your suggestions and recommendations. As suggested, we provided comparative turnaround time for mNGS and histopathologic examination in **line 267** "in our cases, it takes 24 hours for mNGS getting results, 7-10 days for histopathology section preparation and special staining, and about 3 days for common bacteria, 5-7 days for *Nocardia*, and 21 days for *Mycobacterium* in tissue culture". In addition, we added the culture positive rate of these cases in **line 266** " in our cases, the culture positivity rate was only 16.7%". Thanks for your consideration and suggestions, and we believe the revised manuscript would be more improved according to your suggestion.

5. Additional details on the mNGS method should be provided. How was the nucleic acid extracted and how were libraries prepared? What sequencing platform was employed? What was the average depth of sequencing? If this has been published elsewhere, the method could be referenced.

Answer: Thank you for your suggestions and recommendations. As suggested, we have added specific processes and information on nucleic acid extraction, library production, sequencing depth and sequencing platforms: nucleic acid extraction: DNA was isolated using magnetic bead adsorption after lung tissue samples were forcibly fractured with a wall breaker and then treated with lysozyme for 10 min. The BGI metagenomics library kit was used to prepare the library after lung tissue samples were split up by enzymatic digestion, end repair, primer adding, and PCR amplification. In this investigation, the sequencing depth was >20M reads using the MGISEQ2000 sequencing technology with SE50 (**Line 106**). Thanks for your consideration and suggestions, and we believe the revised manuscript would be more improved according to your suggestion.

Minor suggestions:

1. Line 30; the phrase "punctured lesion tissue" may not be familiar to all readers, I might suggest "core needle biopsy tissue" or some other surrogate descriptor.

Answer: Thank you for your suggestions and recommendations. After careful consideration, we thought it was more appropriate to use " core needle biopsy tissue " in **Line 30**, and we have corrected it in the revised manuscript. By reviewing the literature, we found a number of high-quality articles that also used the terminology you suggested [1,2]. Of course, we have corrected similar statements in the other parts of revised manuscript. Thanks for your consideration and suggestions, and we believe the revised manuscript would be more improved according to your suggestion.

References:

1. Richard V, Davey MG, Annuk H, Miller N, Kerin MJ. The double agents in liquid biopsy: promoter and informant biomarkers of early metastases in breast cancer. *Molecular cancer* 2022;21:95.
2. Deleersnijder D, Callemeyn J, Arijs I, et al. Current Methodological Challenges of Single-Cell and Single-Nucleus RNA-Sequencing in Glomerular Diseases. *Journal of the American Society of Nephrology : JASN* 2021;32:1838-52.

2. Line 55; it may be helpful to quantify the positive rate of culture (described as 'low') and provide a reference to help contextualize the results for the reader.

Answer: Thank you for your suggestions and recommendations. As suggested, we reviewed the literature to find specific quantitative values for the positive rate of culture. Therefore, the sentences "the positive rate is low" was modified to "the positive rate is about 27%-40%[3, 4]" (**Line 65**). Thanks for your consideration and suggestions, and we believe the revised manuscript would be more improved according to your suggestion.

3. Line 79; The construction of the sentence beginning "The preliminary clinical diagnosis..." Is somewhat awkward, I would suggest revising the sentence to make the key points more clear.

Answer: Thank you for your suggestions and recommendations. As suggested, our key point is to emphasize the inclusion of conditions that patients should have. Therefore, the sentences

“The preliminary clinical diagnosis of 180 patients may be due to special pathogen infection of the lung, whose lung lesion tissues were tested for mNGS simultaneously.” was modified to “180 patients whose lung lesions were considered to be caused by specific pathogenic microbial infections but required further puncture examination and who simultaneously had their core needle biopsy tissue examined by mNGS were included in the study.” **(Line 88)**. Thanks for your consideration and suggestions, and we believe the revised manuscript would be more improved according to your suggestion.

4. Line 93; A 'lyophilized tube containing 10% formalin' is referenced; this may simply be out of my experience, but I have not encountered 10% formalin in a non-liquid preparation (in our practice is typically buffered formalin in water). Is the descriptor lyophilized appropriate here?

Answer: Thank you for your suggestions and recommendations. As suggested, what we want to express is to put a piece of tissue into a tube with buffered 10% formalin in water. Therefore, the sentences was modified to “In short, when a patient undergoes a percutaneous lung puncture, the operator takes two pieces of lesioned lung tissue of approximately equal length under real-time CT guidance and stores one piece in buffered 10% formalin in water for histopathological testing and one piece in a dry, sterile tube for mNGS testing.” **(Line 101 to 104)**. Thanks for your consideration and suggestions, and we believe the revised manuscript would be more improved according to your suggestion.

5. Line 104; the stain descriptors "hexamine silver staining, and antacid staining" are less familiar to me and may not be readily understood by all readers. Would these stains be equivalent to Gomori-Grocott methenamine silver stain (GMS) and/or Ziehl-Neelsen acid-fast bacilli staining (AFB)?

Answer: Thank you for your suggestions and recommendations. After careful consideration, we thought these stains are equivalent to Gomori-Grocott methenamine silver stain (GMS) and/or Ziehl-Neelsen acid-fast bacilli staining (AFB). By reviewing the literature, we found that Gomori-Grocott methenamine silver stain (GMS) and/or Ziehl-Neelsen acid-fast bacilli stain (AFB) are more commonly used in the literature [1-4] and we have corrected it in the

revised manuscript (**Line 119**). Of course, we have corrected similar statements in the other parts of revised manuscript. Thanks for your consideration and suggestions, and we believe the revised manuscript would be more improved according to your suggestion.

References:

1. Requena L, Sitthinamsuwan P, Santonja C, et al. Cutaneous and mucosal mucormycosis mimicking pancreatic panniculitis and gouty panniculitis. *Journal of the American Academy of Dermatology* 2012;66:975-84.
2. Lavorato FG, Guimarães DA, Premazzi MG, Piñeiro-Maccira JM, Bernardes-Engemann AR, Orofino-Costa R. Performance of mycology and histopathology tests for the diagnosis of toenail onychomycosis due to filamentous fungi: Dermatophyte and non-dermatophyte moulds. *Mycoses* 2017;60:587-93.
3. Lindeboom JA, Bruijnesteijn van Coppenraet LE, van Soolingen D, Prins JM, Kuijper EJ. Clinical manifestations, diagnosis, and treatment of *Mycobacterium haemophilum* infections. *Clinical microbiology reviews* 2011;24:701-17.
4. Franco-Paredes C, Marcos LA, Henao-Martínez AF, et al. Cutaneous Mycobacterial Infections. *Clinical microbiology reviews* 2018;32.

6. Microorganism genus and species names should be italicized.

Answer: Thank you for your suggestions and recommendations. As suggested, we have modified microorganism genus and species names mentioned in the text to the italicized form. Thanks for your consideration and suggestions, and we believe the revised manuscript would be more improved according to your suggestion.

7. Line 125; the descriptor "bidirectional fungi" may be less familiar than "dimorphic fungi".

Answer: Thank you for your suggestions and recommendations. By reviewing the literature, we found a number of high-quality articles that also used the terminology you suggested [1,2]. After careful consideration, we thought it was more appropriate to use "*dimorphic fungi*" in **Line 141**, and we have corrected it in the revised manuscript. Of course, we have corrected similar statements in the other parts of revised manuscript. Thanks for your consideration and

suggestions, and we believe the revised manuscript would be more improved according to your suggestion.

References:

1. Donovan FM, Shubitz L, Powell D, Orbach M, Frelinger J, Galgiani JN. Early Events in Coccidioidomycosis. *Clinical microbiology reviews* 2019;33.
2. Barros MB, de Almeida Paes R, Schubach AO. *Sporothrix schenckii* and Sporotrichosis. *Clinical microbiology reviews* 2011;24:633-54.

8. Line 125; "*Penicillium marneffeii*" was renamed "*Talaromyces marneffeii*" in 2015; it may be helpful to employ the updated name.

Answer: Thank you for your suggestions and recommendations. As you mentioned, "*Penicillium marneffeii*" has been renamed "*Talaromyces marneffeii*" since 2015[1], and the latter is used for high quality articles [2]. After careful consideration, we thought it was more appropriate to use "*Talaromyces marneffeii*" in **Line 142**, and we have corrected it in the revised manuscript. Of course, we have corrected similar statements in the other parts of revised manuscript. Thanks for your consideration and suggestions, and we believe the revised manuscript would be more improved according to your suggestion.

References:

1. Boyce KJ, Andrianopoulos A. Fungal dimorphism: the switch from hyphae to yeast is a specialized morphogenetic adaptation allowing colonization of a host. *FEMS microbiology reviews* 2015;39:797-811.
2. Wang F, Han R, Chen S. An Overlooked and Underrated Endemic Mycosis-Talaromycosis and the Pathogenic Fungus *Talaromyces marneffeii*. *Clinical microbiology reviews* 2023;36:e0005122.

9. Line 151; the expanded form of G/GM tests should be written before the abbreviations are employed.

Answer: Thank you for your suggestions and recommendations. As suggested, we have added the expanded form of G/GM tests (fungal 1-3-beta-D-glucan assay/ galactomannan test) before the abbreviations (**Line 170**). Thanks for your consideration and suggestions, and we

believe the revised manuscript would be more improved according to your suggestion.

10. Line 159; I assume the P-value does not actually equal 0.000, perhaps a less than symbol be employed?

Answer: Thank you for your suggestions and recommendations. As suggested, we have modified the sentence “In addition, the positive rate of mNGS in the infected group was significantly higher than that in the noninfected group (P value < 0.01).” were added and highlighted with red font in **lines 178 to 179** on Page 9 of “Laboratory Test Results” in “Results” section. Thanks for your consideration and suggestions, and we believe the revised manuscript would be more improved according to your suggestion.

11. Line 197; was a beta-lactamase inhibitor used in combination with penicillin as stated, or another beta-lactam antibiotic (such as ampicillin)?

Answer: Thank you for your suggestions and recommendations. The patient was administered with piperacillin sodium tazobactam sodium. After careful consideration, we have modified the sentence “After taking a combination of β -lactamase inhibitors and penicillin antibacterial medications, the patient's cough and hemoptysis improved compared to before.” to “After administered with piperacillin sodium tazobactam sodium, the patient's cough and hemoptysis improved compared to before.” (**Line 220**). Thanks for your consideration and suggestions, and we believe the revised manuscript would be more improved according to your suggestion.

Reviewer #3

Major issues:

- Culture is often not ordered when obtaining a lung biopsy and an infectious process is only suspected after pathology has been performed - comparison of the culture data with the mNGS results would be useful to assess performance. Culture remains the diagnostic standard, from the patients enrolled, how many specimens were actually processed for culture?

Answer: Thank you for your suggestions and recommendations. As suggested, we explained in our article in **line 120** that molecular nucleic acids, such as PCR, are used only when morphology suspects the presence of pathogenic bacteria. A total of 42 of the included cases underwent tissue culture, with a positive rate of 16.7% (**Line 266**). Thanks for your consideration and suggestions, and we believe the revised manuscript would be more improved according to your suggestion.

- This manuscript is either proposing mNGS as a diagnostic tool or a case report - see Case studies section - consider publishing those separately. This manuscript could be more focused

Answer: Thank you for your suggestions and recommendations. This is a diagnostic-related article. And the reason for presenting two specific cases in the article is to better illustrate that mNGS can more accurately reflect the microbial infection of a lesion. There are journals that have published such articles [1]. Thanks for your consideration and suggestions, and we believe the revised manuscript would be more improved according to your suggestion.

Reference:

1. Chen X, Cao K, Wei Y, et al. Metagenomic next-generation sequencing in the diagnosis of severe pneumonias caused by *Chlamydia psittaci*. *Infection* 2020;48:535-42.

- Pathogen abundance in the mNGS data is missing, how significance was assessed in the case of polymicrobial findings

Answer: Thank you for your suggestions and recommendations. As suggested, we added the pathogenic genera detected by patient mNGS and the corresponding sequence numbers in **Table S1**. In this article, we assess the significance of mNGS test results in this way (**Line 136**): Consider the presence of a corresponding pulmonary infection when mNGS has the following conditions: 1) Consider the corresponding infection if influenza virus, parainfluenza virus, respiratory syncytial virus, adenovirus, etc., are detected in the patient's sample; 2) Consider the corresponding infection when microorganisms with difficulty breaking the wall are detected, such as *Mycobacterium bovis*, *Nocardia* spp. and fungi, regardless of their sequence number; 3) identify rare pathogens, such as *Yersinia pestis*

bacteria, dimorphic fungi (*Talaromyces marneffeii*, *Histoplasma capsulatum*, spores, etc.), *Chlamydia psittaci* and various parasitic infections, also considered infections; 4) After excluding common colonizing microorganisms of human skin (such as *Propionibacterium acnes*, *Staphylococcus epidermidis*, etc.), if the pathogenic microbial sequences were greater than 100, they were considered infections; 5) Aspiration pneumonia/lung abscess needs to be considered in patients at risk of aspiration or with a clear history of aspiration whose mNGS findings present with multiple common oral colonizing organisms. Thanks for your consideration and suggestions, and we believe the revised manuscript would be more improved according to your suggestion.

- mNGS detailed procedure or reference to a detailed procedure is missing

Answer: Thank you for your suggestions and recommendations. As suggested, we have added specific processes and information on nucleic acid extraction, library production, sequencing depth and sequencing platforms: nucleic acid extraction: DNA was isolated using magnetic bead adsorption after lung tissue samples were forcibly fractured with a wall breaker and then treated with lysozyme for 10 min. The BGI metagenomics library kit was used to prepare the library after lung tissue samples were split up by enzymatic digestion, end repair, primer adding, and PCR amplification. In this investigation, the sequencing depth was >20M reads using the MGISEQ2000 sequencing technology with SE50 (**Line 106**). Thanks for your consideration and suggestions, and we believe the revised manuscript would be more improved according to your suggestion.

Minor issues:

- Line 88 - disease progression? Instead of regression

Answer: Thank you for your suggestions and recommendations. What we wanted to convey was that we also collected information about the progression of the patient's disease, and after careful discussion, we agreed that "progression" would be more appropriate. Besides, this phrase is also used in some high-quality articles [1,2]. As suggested, we have modified the phrase "disease regression" to "disease progression" (**Line 98**). Thanks for your consideration

and suggestions, and we believe the revised manuscript would be more improved according to your suggestion.

References:

1. Nia HT, Munn LL, Jain RK. Physical traits of cancer. *Science* (New York, NY) 2020;370.
2. Winhammar JM, Rowe DB, Henderson RD, Kiernan MC. Assessment of disease progression in motor neuron disease. *The Lancet Neurology* 2005;4:229-38.

- Scientific names need to be reviewed

o Line 125 *Penicillium marneffeii* is now known as *Talaromyces marneffeii*

Answer: Thank you for your suggestions and recommendations. After careful consideration, we thought it was more appropriate to use "*Talaromyces marneffeii*" in **Line 142**, and we have corrected it in the revised manuscript. As you mentioned, "*Penicillium marneffeii*" has been renamed "*Talaromyces marneffeii*" since 2015[1], and the latter is used for high quality articles [2]. Of course, we have corrected similar statements in the other parts of revised manuscript. Thanks for your consideration and suggestions, and we believe the revised manuscript would be more improved according to your suggestion.

References:

1. Boyce KJ, Andrianopoulos A. Fungal dimorphism: the switch from hyphae to yeast is a specialized morphogenetic adaptation allowing colonization of a host. *FEMS microbiology reviews* 2015;39:797-811.
2. Wang F, Han R, Chen S. An Overlooked and Underrated Endemic Mycosis-Talaromycosis and the Pathogenic Fungus *Talaromyces marneffeii*. *Clinical microbiology reviews* 2023;36:e0005122.

o Line 125? Podocystis - do not know of a clinically relevant fungi named Podocystis histolytica - please review and correct

Answer: Thank you for your suggestions and recommendations. After reviewing literature, we think we used the wrong description, and the clinically relevant fungi should be correctly named *Histoplasma capsulatum* [1,2] (**Line 142**). Of course, we have corrected similar

statements in the other parts of revised manuscript. Thanks for your consideration and suggestions, and we believe the revised manuscript would be more improved according to your suggestion.

References:

1. Bradsher RW. Histoplasmosis and blastomycosis. Clinical infectious diseases : an official publication of the Infectious Diseases Society of America 1996;22 Suppl 2:S102-11.
2. Magrini V, Goldman WE. Molecular mycology: a genetic toolbox for *Histoplasma capsulatum*. Trends in microbiology 2001;9:541-6.

o Line 158 *Klebsiella pneumoniae*

Answer: Thank you for your suggestions and recommendations. After reviewing literature, we found that both "*Klebsiella pneumoniae*" [1,2] and "*Klebsiella pneumonia*" [3,4] can describe this species, but the latter is easily confused with pneumonia caused by this bacterium. Therefore, we used the term "*Klebsiella pneumoniae*" you suggested (**Line 176**). Of course, we have corrected similar statements in the other parts of revised manuscript. Thanks for your consideration and suggestions, and we believe the revised manuscript would be more improved according to your suggestion.

References:

1. Choby JE, Howard-Anderson J, Weiss DS. Hypervirulent *Klebsiella pneumoniae* - clinical and molecular perspectives. Journal of internal medicine 2020;287:283-300.
2. Singh SR, Teo AKJ, Prem K, et al. Epidemiology of Extended-Spectrum Beta-Lactamase and Carbapenemase-Producing Enterobacterales in the Greater Mekong Subregion: A Systematic-Review and Meta-Analysis of Risk Factors Associated With Extended-Spectrum Beta-Lactamase and Carbapenemase Isolation. Frontiers in microbiology 2021;12:695027.
3. Kumar MS, Ghosh S, Nayak S, Das AP. Recent advances in biosensor based diagnosis of urinary tract infection. Biosensors & bioelectronics 2016;80:497-510.
4. de Smalen AW, Ghorab H, Abd El Ghany M, Hill-Cawthorne GA. Refugees and antimicrobial resistance: A systematic review. Travel medicine and infectious disease 2017;15:23-8.

o Line 197 - what is *Clostridium forcestorum* spp?

Answer: Thank you for your suggestions and recommendations. We rechecked the patient's mNGS results and reviewed the literature on the bacterium. The correct expression of this bacterium should be "*Tannerella forsythia*" [1,2] **(Line 219)**. *Tannerella forsythia*, a gram-negative anaerobic bacillus, is one of the important causative agents of periodontal disease and is closely associated with the development of periodontal disease [3]. Of course, we have corrected similar statements in the other parts of revised manuscript. Thanks for your consideration and suggestions, and we believe the revised manuscript would be more improved according to your suggestion.

References:

1. Sharma A. Virulence mechanisms of *Tannerella forsythia*. *Periodontology* 2000 2010;54:106-16.
2. Darveau RP. Periodontitis: a polymicrobial disruption of host homeostasis. *Nature reviews Microbiology* 2010;8:481-90.
3. Tanner AC, Izard J. *Tannerella forsythia*, a periodontal pathogen entering the genomic era. *Periodontology* 2000 2006;42:88-113.

o Line 208 - what is *Microsomonas*

Answer: Thank you for your suggestions and recommendations. We rechecked the patient's mNGS results and reviewed the literature on the bacterium. The correct expression of this bacterium should be "*Parvimonas spp*" [1,2] **(Line 229)**. Of course, we have corrected similar statements in the other parts of revised manuscript. Thanks for your consideration and suggestions, and we believe the revised manuscript would be more improved according to your suggestion.

References:

1. Siqueira JF, Jr., Rôças IN. Microbiology and treatment of acute apical abscesses. *Clinical microbiology reviews* 2013;26:255-73.
2. Kabwe M, Dashper S, Bachrach G, Tucci J. Bacteriophage manipulation of the microbiome associated with tumour microenvironments-can this improve cancer therapeutic

response? FEMS microbiology reviews 2021;45.

- Line 146 - imaging findings? Instead of performance

Answer: Thank you for your suggestions and recommendations. After careful consideration, we thought it was more appropriate to use "imaging findings" in **Line 164**, and we have corrected it in the revised manuscript. Of course, we have corrected similar statements in the other parts of revised manuscript. Thanks for your consideration and suggestions, and we believe the revised manuscript would be more improved according to your suggestion.

- Review paragraph line 163 - special stains in histopathology can direct clinician to potential diagnostics but not detect pathogens: pathology would find acid fast bacilli resembling Mycobacterium or fungal elements/hyphae resembling Aspergillus, but cannot name the pathogens as implied on this statement

Answer: Thank you for your suggestions and recommendations. As suggested, molecular nucleic acid testing will be added after finding pathogenic bacteria in pathological tissue, such as TB-DNA for Mycobacterium, and molecular PCR for Aspergillus and Nocardia to identify pathogens. Therefore, "histopathological testing alone" may confuse the reader, and we changed "histopathological testing alone" to "pathological tissue + molecular nucleic acid testing (PCR)"(**Line 181**). In addition, we modified the means of histopathological examination in **line 120**. Besides, we have removed Figure 2 in response to your comments and replaced it with a more informative **Table S1**. Thanks for your professional consideration and suggestions, and we believe the revised manuscript would be more improved according to your suggestion.

- Line 174 - abbreviations should be defined first time used – ROC

Answer: Thank you for your suggestions and recommendations. As suggested, we have added the expanded form (receiver operating characteristic curve) of "ROC" before the abbreviations when it was used for the first time (**Line 34**). Thanks for your consideration and suggestions, and we believe the revised manuscript would be more improved according to your suggestion.

- Line 238 - *Penicillium* spp. is usually a contaminant not a rare pathogen

Answer: Thank you for your suggestions and recommendations. The reason for mentioning *Penicillium* spp. in the text is that *Talaromyces marneffe* used to belong to the genus *Penicillium*. However, by reviewing the literature, we found that *Talaromyces marneffe* has been classified as a genus of *Talaromyce* [1,2]. Therefore, we changed "*Penicillium* spp." to "*Talaromyce* spp."(**Line 259**). Thanks for your consideration and suggestions, and we believe the revised manuscript would be more improved according to your suggestion.

References:

1. Houbraken J, de Vries RP, Samson RA. Modern taxonomy of biotechnologically important *Aspergillus* and *Penicillium* species. *Advances in applied microbiology* 2014;86:199-249.
2. Limper AH, Adenis A, Le T, Harrison TS. Fungal infections in HIV/AIDS. *The Lancet Infectious diseases* 2017;17:e334-e43.

August 11, 2023

Prof. Dan Liu
Sichuan University West China Hospital
No.37 Guoxue Lane
Chengdu,Sichuan,China
China

Re: Spectrum00586-23R1 (The joint application of metagenomic next-generation sequencing and histopathological examination for the diagnosis of pulmonary infectious disease)

Dear Prof. Dan Liu:

Thank you for submitting the revised manuscript and the revisions have strengthened the study. If possible, please consider submitting the original mNGS data to NCBI to allow for broad availability.

Your manuscript has been accepted, and I am forwarding it to the ASM Journals Department for publication. You will be notified when your proofs are ready to be viewed.

Sincerely,

Arryn Craney
Editor, Microbiology Spectrum
